# Can Language Models Understand Physical Concepts?

**Lei Li[1], Jingjing Xu[2], Qingxiu Dong[3], Ce Zheng[3], Xu Sun[3], Lingpeng Kong[1], Qi Liu[1]**

[1]The University of Hong Kong    [2]Shanghai AI Lab

[3]Peking University

nlp.lilei@gmail.com    {jingjingxu,zce1112zslx,xusun}@pku.edu.cn

dqx@stu.pku.edu.cn    {lpk, liuqi}@cs.hku.hk

## Abstract

Language models (LMs) gradually become general-purpose interfaces in the interactive and embodied world, where the understanding of physical concepts is an essential prerequisite. However, it is unclear whether LMs can understand physical concepts in the human world. To investigate this, we design a benchmark VEC that covers the tasks of (i) **V**isual concepts, such as the shape and material of objects, and (ii) **E**mbodied **C**oncepts, learned from the interaction with the world such as the temperature of objects. Our zero (few)-shot prompting results show that the understanding of certain visual concepts emerges as scaling up LMs, but there are still basic concepts to which the scaling law does not apply. For example, OPT-175B performs close to humans with a zero-shot accuracy of $85\%$ on the material concept, yet behaves like random guessing on the mass concept. Instead, vision-augmented LMs such as CLIP and BLIP achieve a human-level understanding of embodied concepts. Analysis indicates that the rich semantics in visual representation can serve as a valuable source of embodied knowledge. Inspired by this, we propose a distillation method to transfer embodied knowledge from VLMs to LMs, achieving performance gain comparable with that by scaling up parameters of LMs $134\times$.[1]

## 1 Introduction

With the emergent capabilities such as arithmetic (Brown et al., 2020; Wei et al., 2022) and multi-step reasoning (Chowdhery et al., 2022) brought by large-scale pre-training, language models (LMs) are gradually becoming unified interfaces (Hao et al., 2022), capable of instructing embodied robots for high-level tasks such as *cleaning the spilled coke* in interactive and embodied environments (Ahn et al., 2022). Understanding physical concepts is an essential prerequisite for

these tasks, e.g., producing correct instructions for cleaning the coke requires understanding the visual characteristics of a coke can, as well as physical properties such as hardness. However, it still remains unclear whether current LMs can understand basic physical concepts (Driess et al., 2023).

To answer the question, we first define an evaluation suite of physical concepts covering visual and embodied concepts. Specifically, *visual concepts* examine knowledge that can be gained via visual perception, including generic visual concepts, such as color, shape, and material of common objects, and spatial perception, which focuses on the relationship between visual stimuli, i.e., relative size and height of objects. The ability to deal with visual concepts serves as the basis for understanding real-world scenes to perform further instruction. *Embodied concepts* examine knowledge that requires more interaction and multimodal sensory experience in the embodied world, including knowledge about the mass, temperature, and hardness of objects, e.g., ice is colder than water. Understanding embodied concepts is essential for an embodied agent to make correct choices when translating language into actions (Bisk et al., 2020a). We compose a Visual and Embodied Concepts evaluation benchmark **VEC**, with examples shown in Table 1.

With the benchmark, we examine a wide range of LMs. We cover masked language models and causal language models in text-only LMs, including BERT (Devlin et al., 2019) and RoBERTa (Liu et al., 2019b), GPT (OPT)-family (Radford et al., 2019; Zhang et al., 2022b) with parameters ranging from 125M to 175B, LLaMA-1/2 (Touvron et al., 2023a,b) and Vicuna (Chiang et al., 2023). Furthermore, as humans understand the world by learning from multiple modalities, especially using the visual modality (Bloom, 2002), we are interested in whether the vision supervision in recent vision-augmented language models (VLMs) (Chen et al., 2019; Radford et al., 2021; Wang et al., 2022;

---

[1]Our dataset is available at https://github.com/TobiasLee/VEC.

| Concpet Category | | Instance | Label | # of Examples |
|---|---|---|---|---|
| Visual Concepts | Color | $h$: **melon**, $t_1$: **green**, $t_2$: **black** | green | 574 |
| | Shape | $h$: **lemon**, $t_1$: **triangle**, $t_2$: **round** | round | 140 |
| | Material | $h$: **guitar**, $t_1$: **wood**, $t_2$: **glass** | wood | 284 |
| | Size | $h$: **ant**, $r$: **larger than**, $t$: **bird** | false | 500 |
| | Height | $h$: **bottle**, $r$: **shorter than**, $t$: **truck** | true | 500 |
| Embodied Concepts | Mass | $h$: **wooden spoon**, $r$: **heavier than**, $t$: **toaster** | false | 654 |
| | Temperature | $h$: **ice**, $r$: **colder than**, $t$: **water** | true | 422 |
| | Hardness | $h$: **pearl**, $r$: **softer than**, $t$: **glass** | true | 1,016 |

Table 1: The illustration of VEC benchmark. We design two forms of probing tasks. The former (Color, Shape and Material) asks models to make a choice between two tail options given the head object. The latter (Size, Height, and all embodied concepts) requires LMs to judge whether the relation is valid given the head and the tail.

Madureira, 2021) could also facilitate the understanding ability of embodied concepts. CLIP (Radford et al., 2021) and BLIP (Li et al., 2022a) are chosen as representatives of VLMs for evaluation, due to their promising performance and the ability to deal with textual-only inputs. To eliminate the effects of training corpus (Tan and Bansal, 2020), we train BERT, OPT, and CLIP on the same caption dataset with a similar Transformer model (Vaswani et al., 2017) from scratch for a fair evaluation. Furthermore, as previous studies have shown that prompting methods that fit the pre-training paradigm could better elicit the knowledge learned from LMs (Petroni et al., 2019; Schick and Schütze, 2021a; Brown et al., 2020), we adopt pre-trained-objective style promoting methods to narrow the gap between probing and pre-training.

Our zero (few)-shot results on the VEC benchmark show that: (i) Moderate-sized LMs such as BERT and RoBERTa exhibit a random-level understanding of both visual and embodied concepts. (ii) A decent visual understanding of specific concepts emerges as LMs scale up, while they still struggle to understand the embodied knowledge with performance slightly better than random guessing. (iii) Image-grounded caption text-only pre-training, instruction tuning, and visual supervision could provide performance gain regarding visual concepts, yet only the last one enhances the understanding of embodied knowledge of LMs.

We further investigate the source of embodied knowledge in VLMs. A case study demonstrates that embodied knowledge in the VLM of CLIP is potentially rooted in the rich semantics of image representations. We thus devise a knowledge distillation method to transfer the learned embodied knowledge in VLMs into LMs, resulting in an average accuracy gain of 3.38, comparable to the 4.46 gain achieved by scaling the model parameters 134x. Nevertheless, the improved LMs still exhibit great gaps with humans, indicating great potential for further advancements.

## 2  VEC Benchmark

Our VEC benchmark aims to evaluate the understanding of physical concepts of LMs. Inspired by the world scope definitions by Bisk et al. (2020a), we divide physical knowledge into visual knowledge and embodied knowledge. The former are visual properties that can be acquired via visual perception, while the latter focus on knowledge that requires multimodal sensory interaction.

### 2.1  Visual Concepts

Perception is necessary for language learning because it forms the basis for many of our semantic axioms (Bisk et al., 2020a). Among the various types of perception, visual concepts model a vastness of experiences in the world that cannot be stated by text alone (Harnad, 1990). In this work, we consider evaluating the visual understanding ability of LMs by examining their performance on various visual concepts. Specifically, we combine the recently proposed visual knowledge probing datasets, including Spatial Commonsense (Liu et al., 2022a) and ViComTe (Zhang et al., 2022a). The combined dataset requires not only understanding various generic visual concepts including color, shape, and material, but also understanding the relationship between common objects, such as size and height. For generic visual concepts, i.e., color, shape, and material identification, we define an answer selection game: selecting a correct value from two options for the attribute given an object. For

example, given a head object `banana`, the model should pick the ground-truth tail answer `yellow` instead of an alternative option such as `black`. For visual relationships, i.e., size and height understanding, we define a comparison game: LMs need to perform a comparison between different objects. For example, given a head entity `ant` and a tail entity `bird`, the LM is asked to compare the size of two objects and makes a prediction between the correct relation `smaller` and the false one `larger`.

## 2.2 Embodied Concepts

The embodied concepts refer to physical realities of objects, e.g., mass, and temperature, which infants could learn by interacting with the environment (Gopnik et al., 1999). This kind of knowledge is the basis of intelligence and enables agent models to explore challenging tasks in physical environments. We are curious about whether current LMs can capture embodied knowledge via large-scale pre-training. In this work, we define embodied knowledge as the knowledge that requires multimodal sensory interaction with the environments beyond visual perception. We construct embodied knowledge evaluation datasets regarding basic physical properties including mass, temperature, and hardness.

**Mass Dataset**    We build the Mass dataset by transforming the Image2Mass dataset curated by Standley et al. (2017), which annotates 56 common objects with corresponding weights. The most light-weight object in the dataset is a red Lego brick, weighing 0.026 lbs, and the heaviest object is a 2.664 lbs drill. Directly asking the LM for the absolute mass of objects can be challenging (Wallace et al., 2019). We define the task in a comparison format. Specifically, each comparison pair contains two objects with a weight gap greater than 1 lbs. The threshold is set according to the Weber–Fechner laws (Fechner et al., 1966) to guarantee that the mass difference is perceivable for humans. We build 654 triplets such as (`hair dryer, heavier than, red Lego brick`) for evaluation.

**Temperature Dataset**    We design a temperature probing dataset by collecting the temperature of 22 common objects from Wikipedia.[2] For example, the ice is 0°C, and the temperature of water

vapor is 100°C. We convert the object with temperature annotations into pairs, and each pair contains two objects and the corresponding temperature relation. For example, (`ice, colder than, water vapor`). The temperature gap between two objects must be greater than a difference threshold, which is loosely set to 10°C for assurance of thermal perception for human (Jones, 2009). The final Temperature dataset consists of 422 pairs in total.

**Hardness Dataset**    Hardness is a measure of the resistance to localized plastic deformation in material science. For example, hard metals such as titanium are harder than soft minerals such as talc. Humans can perceive the hardness of different materials in interaction with the environment by using tactile organs like fingers (Gueorguiev et al., 2016). To investigate whether LMs capture hardness knowledge, we build a Hardness dataset by collecting the Mohs hardness scores of 25 objects from Wikipedia.[3] We define the task in a comparison format. For example, (`talc, softer than, titanium`). Each pair contains two objects. The gap between two objects is greater than the threshold for human-level understanding. The final dataset contains 1,016 pairs.

## 3   Prompting Methods

Recent studies have shown that prompting methods that fit the pre-training paradigm are more effective than other possible prompting methods (Petroni et al., 2019; Schick and Schütze, 2021a). Following these studies, we design specific prompts for LMs with different objectives.

**Prompting Masked Language Models**    Following PET (Schick and Schütze, 2021a,b), we probe the masked language models by converting knowledge facts into a question-answering form. For example, a size knowledge fact (`coin, smaller than, table`) is converted into a sentence with a special mask token: `Question: is a coin smaller than a table? Answer: [MASK]`. We also explored other prompts, such as `Is a coin [MASK] than table`. However, our experiments show that a question-answering form can better induce models to generate answers and avoid the influence of tokenization of different LMs. Given masked inputs, the model is asked to predict the probabilities of the mask token over two choices,

---

[2]`https://en.wikipedia.org/wiki/Orders_of_magnitude_(temperature)`

[3]`https://en.wikipedia.org/wiki/Mohs_scale_of_mineral_hardness`

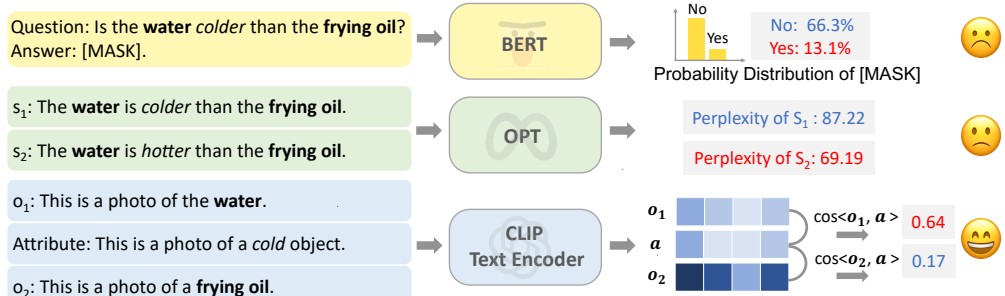

Figure 1: An illustration of prompting methods. For BERT-like models with a masked language head, we convert the knowledge fact to a question and perform prediction with the head over *yes* or *no*. For OPT models, we evaluate the perplexity of different assertions and take the one with lower perplexity as a valid fact. For CLIP, we devise a matching-based probing framework.

i.e., `yes` for confirming the knowledge fact is valid or `no` for an unreasonable assertion. We observe that in specific LMs, the prediction can be biased toward some answers as investigated by Zhao et al. (2021). We calibrate the prediction by normalizing the probabilities according to an estimated prior following Zhao et al. (2021).

**Prompting Causal Language Models** Different from BERT, there is no special [MASK] token in causal language models like GPT (Radford et al., 2019). Therefore, introducing a special token would result in an inconsistency between pre-training and evaluation. To remedy this, for each knowledge fact, we state it in natural sentences according to prompting templates and evaluate the sentence perplexity as the proxy metric. Specifically, for size-property evaluation, we convert it into a valid knowledge assertion $s1 = $ `A coin is smaller than a table`, and an invalid one by replacing the relation with the antonym adjective $s2 = $ `A coin is larger than a table`. The sentence with lower perplexity is then chosen as the predicted one. We evaluate the perplexity of each sentence $s = (w_0, w_1, \cdots, w_n)$ as:

$$\text{PPL}(s) = P_{\mathcal{M}}(s)^{-\frac{1}{n}} = \sqrt[n]{\prod_{k=1}^{n} \frac{1}{P_{\mathcal{M}}(w_k \mid w_0, w_1, \ldots, w_{k-1})}}$$

where $P_{\mathcal{M}}$ denotes the conditional word probability of the causal language model to be probed and $n$ is the number of tokens in $s$. We compare the perplexity $\text{PPL}(s_1)$ and $\text{PPL}(s_2)$ and choose the sentence with lower PPL as a more valid assertion and calculate the prediction accuracy accordingly.

**Prompting Vision-augmented Language Models of CLIP** Unlike text-only LMs that support word predictions, the text encoder in CLIP only has one

sentence representation without any pre-trained language heads. To probe the learned knowledge in VLMs such as CLIP, we design a matching-based prompting method. In more detail, for the size fact stated before, we first obtain two object descriptions $o_1 = $ `a photo of a coin`, and $o_2 = $ `a photo of a table`. These two sentences are encoded to get the corresponding object vectors via the CLIP language encoder:

$$\mathbf{o}_1, \mathbf{o}_2 = \text{CLIP}(o_1), \text{CLIP}(o_2).$$

We then derive an attribute sentence $a = $ `a photo of a small object`, and encode it to an attribute adjective vector with the language encoder:

$$\mathbf{a} = \text{CLIP}(a).$$

The prediction is then performed by comparing the cosine similarity $\cos(\mathbf{o}_1, \mathbf{a})$ and $\cos(\mathbf{o}_2, \mathbf{a})$.[4] The object with higher similarity with the attribute description is adopted as the answer, i.e., a coin is smaller than a table, if $\cos(\mathbf{o}_1, \mathbf{a}) > \cos(\mathbf{o}_2, \mathbf{a})$. Otherwise, we assume that the model thinks the reversed relation holds. We can also adopt the antonym adjective *large* for getting the attribute vectors. The results of the best-performing adjective words for CLIP are reported and we discuss the influence of adjective options in § 4.3.

## 4 Experiments

### 4.1 Experimental Settings

**Models** We cover two kinds of LMs, text-only LMs and visual-augmented LMs. Text-only LMs include BERT-base/large (Devlin et al., 2019),

---

[4]The matching-based prompting also applies to the pooled embedding of BERT, yet the results exhibit great variance as shown in Appendix A.

| Model (# of Param.) | Color | Shape | Size | Height | Material | Average |
|---|---|---|---|---|---|---|
| BERT$_{\text{YFCC-15M}}$ (63M) | $56.05_{\pm 10.36}$ | $53.21_{\pm 1.79}$ | $50.34_{\pm 1.27}$ | $50.16_{\pm 1.30}$ | $55.35_{\pm 4.79}$ | 53.02 |
| OPT$_{\text{YFCC-15M}}$(63M) | $65.21_{\pm 15.27}$ | $51.25_{\pm 18.99}$ | $50.50_{\pm 0.77}$ | $49.96_{\pm 1.36}$ | $81.41_{\pm 1.53}$ | 59.67 |
| CLIP$_{\text{YFCC-15M}}$ (63M) | $68.21_{\pm 7.17}$ | $67.21_{\pm 7.63}$ | $62.64_{\pm 6.01}$ | $54.04_{\pm 7.05}$ | $62.92_{\pm 6.48}$ | 63.00 |
| BERT-base (110M) | $49.29_{\pm 1.60}$ | $52.14_{\pm 4.22}$ | $49.94_{\pm 0.80}$ | $50.56_{\pm 0.59}$ | $48.08_{\pm 2.74}$ | 50.00 |
| BERT-large (340M) | $49.36_{\pm 1.88}$ | $51.21_{\pm 5.06}$ | $49.26_{\pm 1.60}$ | $49.08_{\pm 2.34}$ | $49.72_{\pm 0.58}$ | 49.73 |
| RoBERTa-base (125M) | $49.07_{\pm 1.62}$ | $49.36_{\pm 3.52}$ | $50.32_{\pm 0.57}$ | $49.58_{\pm 0.49}$ | $49.86_{\pm 1.44}$ | 49.64 |
| RoBERTa-large (355M) | $49.66_{\pm 0.54}$ | $50.68_{\pm 1.48}$ | $50.54_{\pm 1.46}$ | $50.14_{\pm 0.45}$ | $50.00_{\pm 0.14}$ | 50.20 |
| OPT (125M) | $70.02_{\pm 9.59}$ | $57.32_{\pm 6.46}$ | $45.98_{\pm 4.23}$ | $56.76_{\pm 1.36}$ | $82.43_{\pm 2.20}$ | 62.50 |
| OPT (1.3B ) | $76.92_{\pm 5.97}$ | $65.00_{\pm 6.12}$ | $51.12_{\pm 2.66}$ | $57.82_{\pm 4.46}$ | $85.63_{\pm 3.49}$ | 67.30 |
| OPT (13B) | $79.62_{\pm 5.28}$ | $62.50_{\pm 6.44}$ | $57.56_{\pm 6.60}$ | $54.58_{\pm 4.53}$ | $\mathbf{88.38}_{\pm 3.14}$ | 68.53 |
| OPT (175B) | $\mathbf{83.10}_{\pm 3.13}$ | $65.71_{\pm 7.54}$ | $59.18_{\pm 9.05}$ | $55.84_{\pm 5.33}$ | $85.49_{\pm 2.01}$ | 69.87 |
| LLaMa-1 (7B) | $63.94_{\pm 4.87}$ | $66.19_{\pm 2.36}$ | $65.91_{\pm 9.86}$ | $50.00_{\pm 0.00}$ | $66.76_{\pm 3.88}$ | 62.56 |
| Vicuna-v1.3 (7B) | $64.31_{\pm 5.44}$ | $73.33_{\pm 2.88}$ | $62.50_{\pm 8.80}$ | $50.02_{\pm 0.11}$ | $68.31_{\pm 3.75}$ | 63.69 |
| LLaMa-1 (13B) | $66.27_{\pm 3.89}$ | $62.38_{\pm 2.36}$ | $63.14_{\pm 11.06}$ | $50.16_{\pm 0.43}$ | $65.46_{\pm 2.95}$ | 61.48 |
| Vicuna-v1.3 (13B) | $66.11_{\pm 5.62}$ | $67.38_{\pm 2.69}$ | $64.35_{\pm 13.62}$ | $50.92_{\pm 2.53}$ | $68.52_{\pm 5.60}$ | 63.46 |
| LLaMa-2 (7B) | $63.73_{\pm 3.09}$ | $65.24_{\pm 4.22}$ | $61.88_{\pm 9.37}$ | $50.02_{\pm 0.06}$ | $66.34_{\pm 3.25}$ | 61.44 |
| LLaMa-2-Chat (7B) | $60.99_{\pm 5.18}$ | $70.95_{\pm 2.76}$ | $68.03_{\pm 9.91}$ | $51.72_{\pm 2.20}$ | $67.39_{\pm 4.13}$ | 63.82 |
| LLaMa-2 (13B) | $66.59_{\pm 3.40}$ | $62.38_{\pm 3.21}$ | $68.20_{\pm 11.51}$ | $50.10_{\pm 0.18}$ | $66.73_{\pm 3.99}$ | 62.80 |
| LLaMa-2-Chat (13B) | $64.04_{\pm 4.41}$ | $70.71_{\pm 1.75}$ | $70.68_{\pm 8.79}$ | $51.18_{\pm 1.61}$ | $67.96_{\pm 4.63}$ | 64.91 |
| CLIP-ViT/B-32 (63M) | $80.07_{\pm 2.57}$ | $84.43_{\pm 2.57}$ | $61.40_{\pm 6.02}$ | $62.28_{\pm 6.40}$ | $80.07_{\pm 2.57}$ | 73.94 |
| DeCLIP-ViT/B-32 (63M) | $81.48_{\pm 2.63}$ | $84.07_{\pm 2.34}$ | $\mathbf{76.92}_{\pm 1.81}$ | $68.12_{\pm 2.15}$ | $81.48_{\pm 2.63}$ | 78.35 |
| CLIP-ViT/L-14 (123M) | $80.33_{\pm 3.61}$ | $\mathbf{85.00}_{\pm 4.03}$ | $63.96_{\pm 6.10}$ | $60.72_{\pm 5.56}$ | $80.33_{\pm 3.61}$ | 74.21 |
| BLIP-base (138M) | $82.60_{\pm 5.50}$ | $84.86_{\pm 2.80}$ | $76.00_{\pm 6.40}$ | $\mathbf{69.84}_{\pm 7.76}$ | $80.67_{\pm 1.24}$ | $\mathbf{78.79}$ |

Table 2: Zero-shot probing results on visual datasets. Models with the YFCC-15M subscript represents that these models are trained from scratch on YFCC-15M data. Scaling OPT-family brings clear improvements on size and color datasets. The scaling law fails on the height dataset.

RoBERTa-base/large (Liu et al., 2019b) for masked language models, and OPT models with parameters ranging from 125M to 175B. We further incorporate recent variants of causal language models into evaluation, including LLaMA-1/2 (7B and 13B) (Touvron et al., 2023a), Vicuna models (7B and 13B, v1.3) (Chiang et al., 2023) trained with the instruction tuning dataset, and LLaMa-2 Chat models (7B and 13B) (Touvron et al., 2023b) trained with supervised fine-tuning and RLHF (Ouyang et al., 2022). For VLMs, we include the text encoders of CLIP-ViT-B/32 and CLIP-ViT-L/14 (Radford et al., 2021) as a base and a large version, respectively. We also include an enhanced VLM with masked language modeling as self-supervision, DeCLIP-ViT-B/32 (Li et al., 2022b) and BLIP, a boosted VLM by unifying multi-modal understanding and generation tasks (Li et al., 2022a).[5] Since directly comparing the VLMs and text-only LMs can be unfair due to the difference in model configuration and training corpus (Tan and Bansal, 2020), we re-train CLIP, BERT, and GPT from scratch with a similar Transformer model on the same text corpus, the

caption dataset in the YFCC-15M dataset (Thomee et al., 2016). All models are trained for 32 epochs. The only difference between these models is the pre-training objective. Detailed model and training settings are elaborated in Appendix B.

**Prompts** We manually write several prompts (at least 4 prompts for each task) to eliminate the side-effect of the expression variations and report the averaged accuracy. Besides, the variance across different prompts could also serve as an indicator to evaluate the robustness of learned knowledge facts. We report the averaged performance over multiple prompts for all models. All used prompts can be found in Appendix C.

### 4.2 Main Findings

**The ability of certain visual concepts emerges as scaling up LMs, but there are still basic visual concepts where the scaling law fails.** The evaluation results on visual datasets are shown in Table 2. Interestingly, with the scaling up of OPT-family models, the prediction accuracy increases obviously on specific visual concepts such as color and size. On material and color, the largest OPT-175B model even achieves better results than VLMs of CLIP-ViT/L-14, which are augmented

---

[5] https://huggingface.co/Salesforce/blip-itm-base-coco

| Model (# of Param.) | Mass | Temperature | Hardness | Average |
|---|---|---|---|---|
| BERT$_{\text{YFCC-15M}}$(63M) | $50.73_{\pm 2.53}$ | $49.50_{\pm 1.19}$ | $50.91_{\pm 1.04}$ | 50.38 |
| GPT$_{\text{YFCC-15M}}$(63M) | $50.02_{\pm 0.05}$ | $57.73_{\pm 2.24}$ | $50.04_{\pm 2.98}$ | 52.61 |
| CLIP$_{\text{YFCC-15M}}$(63M) | $67.45_{\pm 5.16}$ | $64.83_{\pm 4.17}$ | $62.22_{\pm 3.11}$ | 64.83 |
| BERT-base (110M) | $50.35_{\pm 0.56}$ | $49.67_{\pm 0.56}$ | $50.20_{\pm 0.43}$ | 50.07 |
| BERT-large (340M) | $49.97_{\pm 1.31}$ | $49.83_{\pm 0.50}$ | $49.98_{\pm 0.06}$ | 49.93 |
| RoBERTa-base (125M) | $49.65_{\pm 0.51}$ | $50.00_{\pm 0.00}$ | $48.04_{\pm 2.04}$ | 49.23 |
| RoBERTa-large (355M) | $50.08_{\pm 0.23}$ | $50.07_{\pm 0.19}$ | $49.95_{\pm 0.15}$ | 50.03 |
| OPT (125M) | $50.00_{\pm 0.00}$ | $54.53_{\pm 4.33}$ | $46.16_{\pm 2.45}$ | 50.23 |
| OPT (1.3B) | $50.05_{\pm 0.10}$ | $50.90_{\pm 5.08}$ | $53.03_{\pm 2.69}$ | 51.33 |
| OPT (13B) | $50.14_{\pm 0.36}$ | $51.85_{\pm 6.34}$ | $52.38_{\pm 3.09}$ | 51.46 |
| OPT (175B) | $50.21_{\pm 0.24}$ | $59.83_{\pm 8.68}$ | $57.33_{\pm 3.41}$ | 55.79 |
| LLaMa-1 (7B) | $54.88_{\pm 2.49}$ | $60.69_{\pm 4.35}$ | $51.97_{\pm 2.84}$ | 55.84 |
| Vicuna-v1.3 (7B) | $54.23_{\pm 1.78}$ | $58.85_{\pm 4.36}$ | $54.42_{\pm 6.42}$ | 55.83 |
| LLaMa-1 (13B) | $53.69_{\pm 3.81}$ | $50.76_{\pm 8.69}$ | $53.94_{\pm 4.45}$ | 52.80 |
| Vicuna-v1.3 (13B) | $56.90_{\pm 3.53}$ | $53.32_{\pm 6.47}$ | $55.50_{\pm 5.73}$ | 55.24 |
| LLaMa-2 (7B) | $54.01_{\pm 4.47}$ | $56.87_{\pm 6.25}$ | $55.22_{\pm 5.89}$ | 55.37 |
| LLaMa-2-Chat (7B) | $52.51_{\pm 4.83}$ | $61.99_{\pm 3.93}$ | $55.65_{\pm 5.28}$ | 56.72 |
| LLaMa-2 (13B) | $53.38_{\pm 2.10}$ | $57.54_{\pm 7.51}$ | $53.01_{\pm 4.57}$ | 54.64 |
| LLaMa-2-Chat (13B) | $54.13_{\pm 2.73}$ | $56.68_{\pm 6.02}$ | $54.12_{\pm 4.64}$ | 54.98 |
| CLIP-ViT/B-32 (63M) | $65.20_{\pm 4.75}$ | $60.28_{\pm 6.83}$ | $59.43_{\pm 2.00}$ | 61.64 |
| DeCLIP-ViT/B-32 (63M) | $54.95_{\pm 2.00}$ | $68.58_{\pm 3.08}$ | $61.10_{\pm 4.14}$ | 61.54 |
| CLIP-ViT/L-14 (123M) | $73.15_{\pm 6.34}$ | $65.88_{\pm 2.31}$ | $\mathbf{69.57}_{\pm 2.26}$ | 69.53 |
| BLIP-base (138M) | $\mathbf{83.94}_{\pm 2.59}$ | $\mathbf{74.98}_{\pm 5.60}$ | $56.93_{\pm 5.56}$ | **71.95** |

Table 3: Zero-shot results on embodied datasets. LMs struggle to understand embodied knowledge, including OPT (175B) and visual-augmented LMs, with 71.95 as the best average performance.

| Model | Mass | Temperature | Hardness | Avg. |
|---|---|---|---|---|
| Zero-shot Best VLMs | $\mathbf{83.94}_{\pm 2.59}$ | $\mathbf{74.98}_{\pm 5.60}$ | $\mathbf{69.57}_{\pm 2.26}$ | **76.16** |
| BERT-base | $64.72_{\pm 4.77}$ | $55.62_{\pm 1.34}$ | $51.80_{\pm 1.31}$ | 57.38 |
| BERT-large | $65.47_{\pm 4.86}$ | $54.19_{\pm 2.31}$ | $52.73_{\pm 1.22}$ | 57.46 |
| RoBERTa-base | $60.24_{\pm 5.24}$ | $60.27_{\pm 4.85}$ | $50.44_{\pm 1.13}$ | 56.98 |
| RoBERTa-large | $61.18_{\pm 4.01}$ | $58.28_{\pm 2.09}$ | $50.56_{\pm 1.14}$ | 56.67 |

Table 4: The few-shot results of BERT variants. With 16 instances, the fine-tuned BERT variants are still worse than zero-shot visual-augmented LMs.

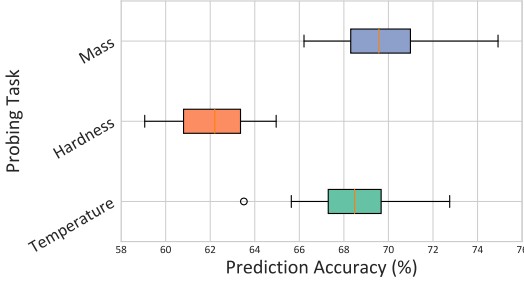

Figure 2: Few-shot results of OPT-175B with 16 instances as demonstration on embodied tasks.

with vision supervision and are supposed to perform better (Zhang et al., 2022a; Liu et al., 2022b). A potential reason is that the combination of color and material frequently occurs (e.g., red apples) in raw texts, and these co-occurrence statistics are well captured by large LMs. The significant performance improvements after training on visual-grounded text corpus YFCC-15M validate this explanation. Besides, OPT-13B and LLaMa-1 13B models excel in different visual concepts, with OPT-13B performing well on material concepts and LLaMa-1 13B on relative size comparisons, likely due to the difference of pre-training corpus distribution. However, increasing LMs to 175B brings negligible improvements in the Height dataset, indicating that there still remain visual concepts where the scaling law does not hold even though these concepts can be easily captured by humans.

**LMs exhibit a poor understanding of embodied concepts.** As shown in Table 3, the scaling law fails again on the embodied concepts, as all LMs, including OPT-175B and variants trained with captions data, perform poorly. Among LMs, the LLaMa series shows a better performance in embodied concepts, yet still reaches a plateau of around 55% overall accuracy. We further conduct a few-shot prompt evaluation for OPT models by constructing the inputs with $k = 16$ randomly sampled instances and adopt PET (Schick and Schütze, 2021a) for masked language models. The results are illustrated in Figure 2 and Table 4, respectively. We find that while the performance is boosted, the average results are still worse than the CLIP-ViT/L-

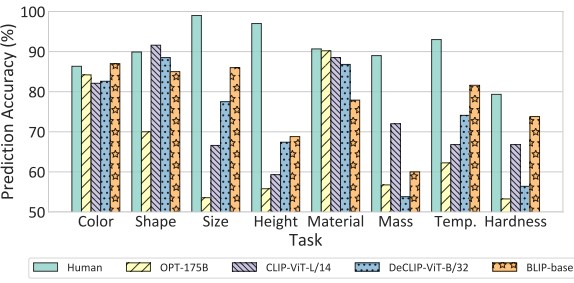

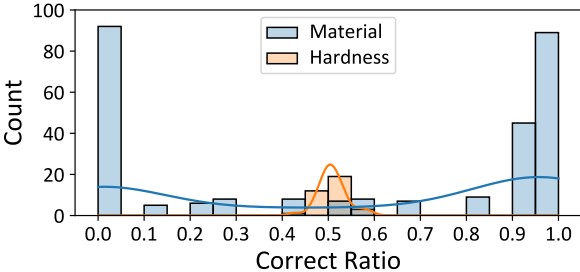

Figure 3: Comparison between the best-performing models and human annotators on sampled subsets of VEC. The best-performing LMs and VLMs achieve close-to-human results on visual datasets, yet far lag behind humans in embodied datasets.

Figure 4: Entity correct ratio histograms of Mass and Material datasets across different prompts. BERT could make consistent correct predictions for specific entities, and the bell curve on the hardness indicates it is challenging for BERT to understand embodied concepts.

14 model without any demonstration, which only utilizes 0.08% parameters of OPT-175B. These findings show that visual supervision can help learn embodied knowledge, but there is still a large gap between the best results of existing LMs with human performance.

**Compared with human annotators, OPT-175B and VLMs achieve competitive performance regarding visual concepts, yet they exhibit great gaps with humans on embodied concepts.** We conduct a human evaluation to better understand the performance of different models. Specifically, we randomly sample 100 examples for each task and ask three volunteers to label these examples. The annotators achieve substantial agreements on all the tasks with Cohen's kappa (Cohen, 1960) $\kappa$ larger than $0.7$, except for the Hardness dataset with a moderate $\kappa = 0.52$. The comparison with best-performing models, i.e., OPT-175B, CLIP-ViT/L-14 and DeCLIP is illustrated in Figure 3. We find that (i) Regarding visual concepts, both OPT and CLIP-like models perform closely to human annotators. CLIP and DeCLIP even outperform the human annotators on the shape task, which is potentially due to the noise introduced by the automatic construction of the dataset (Zhang et al., 2022a). Overall, the close-to-human results indicate that visual knowledge can be effectively acquired by large-scale cross-modal pre-training or even text-only pre-training with sufficient parameters. (ii) Regarding embodied concepts, the best-performing CLIP-ViT-L/14 model still has an absolute $18.5\%$ accuracy gap with the humans. The clear performance gaps reveal that there is still a long way to go in equipping LMs and VLMs with embodied knowledge.

**Instruction tuning enhances proficiency in both visual and embodied concepts.** After post-training with the instruction tuning dataset, Vicuna models display enhanced proficiency in both visual and embodied concepts, with larger LLMs demonstrating a more significant improvement. For instance, when using LLaMa-1 (13B) as a baseline model, the average accuracy in three embodied tasks rises from $52.8$ to $55.2$. Moreover, LLaMa-2-Chat models, which are further trained with a supervised instruction tuning dataset and RLHF techniques, show consistent accuracy gains in both visual and embodied concept tasks as well. However, disentangling the influence of instruction tuning and RLHF on these models presents a challenge as they are intertwined. Nevertheless, a clear performance gap still remains between more recent LMs and VLMs, indicating the significance of visual supervision.

### 4.3 Analysis

**Does BERT behave similarly regarding visual and embodied concepts?** The overall prediction results of BERT-like models in the visual and embodied world are both at a random level. We investigate this question result by first checking whether BERT models perform consistently at a guessing level for all the entities in the dataset. We compute the entity correct ratio among different prompts for the objects in different datasets and compare the distribution on different tasks with the BERT model trained on YFCC-15M dataset. As illustrated in Figure 4, in the Material identification task, there are entities that the model that could provide consistent correct predictions. However, the distribution of the Hardness dataset in embodied evaluation exhibits a bell curve, i.e., most entities are predicted

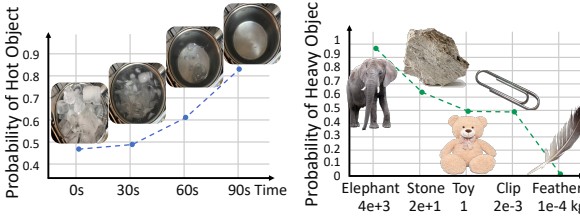

Figure 5: Case study showing that the image representations in CLIP exhibit embodied knowledge. (Left) The probability of an image being classified as "hot" increases as the ice melts being heated in a boiler in a video. (Right) The probability of an image being classified as "heavy" along with corresponding mass annotation.

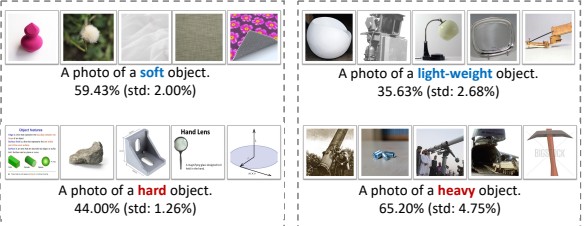

Figure 6: Top-5 retrieved images and the prediction accuracy with different attribute prompts. The accuracy drops when the text inputs contain ambiguous words and compound words, as the retrieved images are biased toward specific meanings.

correctly at a random-chance level. The distributions of other tasks show similar results and can be found in Appendix D. These results suggest that BERT learns visual knowledge for certain entities yet indeed struggles regarding embodied concepts.

**Exploring learned embodied knowledge in image representations.** We are interested in how the VLMs of CLIP learn embodied knowledge. A potential answer is that the images contain rich semantics regarding embodied knowledge such as the heat of the object, and the knowledge can be propagated to the VLMs via the contrastive learning objective. To examine this, we perform a case study by calculating the attribute similarities among the images. We first take clips from a video of heating a pile of ice and then perform a binary classification by calculating the cosine similarities with text prompts *a photo of a hot object.* and *a photo of a cold object* for each frame. The left of Figure 5 shows that the probability of a hot object increases during the heating procedure. Similarly, we perform a binary classification over heavy and light-weight objects ranging from an elephant to a feather. The right of Figure 5 shows that the image representations are aware of the mass of different objects. This qualitative study shows that image representations are the potential source of embodied knowledge.

**Transferring embodied knowledge from VLMs to LMs.** We further verify whether the learned embodied knowledge in CLIP could be transferred to text-only models. Specifically, we perform knowledge distillation (Hinton et al., 2015) by treating the original text-only language model as a student, and the CLIP text encoder as a teacher model providing the learned embodied knowledge. How-

ever, our preliminary study in Appendix F shows that vanilla alignment on the predicted word distributions could not be effective. Inspired by our case study showing that the rich embodied knowledge contained in the representations, we utilize Neuron Selectivity Transfer (Huang and Wang, 2017) which transfers the inner states such as spatial activation patterns of teacher neurons to student neurons, by aligning the token representations of the last layer between the teacher and student language models, which is implemented as a squared maximum mean discrepancy (MMD) with a polynomial kernel to measure the distance between the activation patterns of student neurons and teacher neurons. The total training objective of the language model is a combination of the original language modeling loss and the MMD loss with a balancing coefficient. We refer readers to Appendix E for more details. As shown in Table 5, the distillation provides a performance boost on embodied concepts understanding, e.g., learning from a CLIP-ViT-L/14 teacher model achieves improvement that is comparable with that brought by scaling the model parameter 134x from OPT-1.3B to OPT-175B.[6] It validates our assumption and indicates that future studies could utilize the richer representations in VLMs for improving LMs, yet the gap between distilled LM and VLM suggests that there is still room for advancement.

**VLMs perform poorly when dealing with ambiguous text descriptions.** During our preliminary study, we observe that VLMs of CLIP perform relatively poorly for specific adjectives such as *hard*. To further investigate this issue, we examine the retrieved images using prompts with different attribute adjectives on the CC12M dataset (Chang-

---

[6]Only OPT is adopted for experiments as the CLIP encoder cannot deal with the mask tokens introduced in BERT.

pinyo et al., 2021). Our results, illustrated in Figure 6, revealed that for the prompt *a photo of a hard object*, the retrieved images were mostly about abstract and difficult learning materials, with only one rock image related to the attribute of hardness. Additionally, for the prompt *light-weight*, the retrieved images are biased towards meanings related to lighting bulbs and light-toned colors. These observations demonstrate that handling semantic ambiguity remains a challenge for VLMs (Ren et al., 2023), suggesting that future improvements may incorporate more language-side supervision into the text encoder of VLMs (Li et al., 2022b).

## 5 Related Work

**Probing Language Models**   Understanding what LMs know after large-scale pre-training is an active research area (Rogers et al., 2020). Various probing methods have been developed (Tenney et al., 2019b; Petroni et al., 2019), and investigations show that LMs capture linguistic (Tenney et al., 2019a; Liu et al., 2019a), factual (Petroni et al., 2019; Roberts et al., 2020; Dai et al., 2022), commonsense knowledge (Wang et al., 2019; Forbes et al., 2019), and even acquire grounded concepts (Patel and Pavlick, 2022). For VLMs, studies demonstrate their potential in acquiring spatial commonsense (Zhang et al., 2022a; Liu et al., 2022a; Xia et al., 2023) and color perception (Abdou et al., 2021), yet performing worse on NLU tasks (Tan and Bansal, 2020) and achieving no significant on lexical grounding (Yun et al., 2021). In this paper, we investigate the ability of LMs to understand physical concepts. Different from PIQA (Bisk et al., 2020b) consists of questions requiring physical commonsense reasoning, our VEC benchmark examines the understanding ability of the fundamental physical concepts. The evaluation on the VEC benchmark demonstrates that text-only LMs can learn specific visual concepts after scaling up while struggling with the embodied concepts.

**Vision-Language Pre-training**   Unifying cross-modal representations via vision-language pre-training has achieved promising progress. Pilot studies adopt masked reconstruction to learn shared representations across modalities from a mixed visual and language inputs (Li et al., 2019; Tan and Bansal, 2019; Su et al., 2020; Chen et al., 2019; Li et al., 2020). CLIP (Radford et al., 2021) introduces a contrastive language-image pre-training framework, utilizing language as supervision for

| Model | Mass | Temperature | Hardness | Avg. |
|---|---|---|---|---|
| CLIP-ViT/B-32 (T1) | $65.20_{\pm 4.75}$ | $60.28_{\pm 6.83}$ | $59.43_{\pm 2.00}$ | 61.64 |
| CLIP-ViT/L-14 (T2) | $73.15_{\pm 6.34}$ | $65.88_{\pm 2.31}$ | $69.57_{\pm 2.26}$ | 69.53 |
| OPT-1.3B | $50.05_{\pm 0.10}$ | $50.90_{\pm 5.08}$ | $53.03_{\pm 2.69}$ | 51.33 |
| scale up to 13B | $50.14_{\pm 0.36}$ | $51.85_{\pm 6.34}$ | $52.38_{\pm 3.09}$ | 51.46 (+0.13) |
| scale up to 175B | $50.21_{\pm 0.24}$ | $59.83_{\pm 8.68}$ | $57.33_{\pm 3.41}$ | 55.79 (+4.46) |
| OPT$_{\text{YFCC-15M}}$ | $50.02_{\pm 0.05}$ | $57.73_{\pm 2.24}$ | $50.04_{\pm 2.98}$ | 52.61 |
| Distill w/ T1 | $49.88_{\pm 0.37}$ | $55.76_{\pm 4.01}$ | $\mathbf{53.23}_{\pm 3.12}$ | 52.96 (+0.35) |
| Distill w/ T2 | $\mathbf{54.27}_{\pm 5.20}$ | $\mathbf{60.78}_{\pm 4.23}$ | $52.91_{\pm 1.62}$ | $\mathbf{55.99}$ (+3.38) |

Table 5: Results of embodied distillation. Transferring embodied knowledge from CLIP-ViT to OPT brings a gain of 3.38 points, which is comparable with the improvements by scaling the model from 1.3B to 175B.

learning transferable image representations with large-scale image-text pairs, triggering a series of variants for further improvements (Jia et al., 2021; Li et al., 2022b; Yao et al., 2022; Li et al., 2021, 2022a). Our study uses VLMs of CLIP and BLIP to investigate the impact of visual supervision on understanding physical concepts and our results suggest that visual supervision is crucial for LMs to understand embodied concepts, which can be utilized to enhance the text-ony LMs.

## 6 Conclusion

In this paper, we introduce **VEC** for evaluating the understanding of physical concepts in LMs. Our results show that large LMs understand specific visual concepts but struggle with embodied knowledge. VLMs instead perform much better in both the visual and the embodied world, indicating that visual signals are vital for understanding physical concepts. Further analysis suggests that transferring the VLM representations to LMs effectively boosts embodied concepts understanding, shedding light on directions for improving LMs.

## Limitations

**Limited Scopes of Physical Concepts**   In this work, we focus on evaluating certain physical properties such as color, mass, temperature, and hardness. These properties are chosen because they can be measured using well-established metrics and are easily sensed by humans. However, this selection introduces a bias in our approximation of embodied knowledge. Despite this bias, our results are still sufficient to demonstrate the poor performance of current text-only language models (LMs) in understanding embodied concepts. We suggest that incorporating vision supervision could help improve the understanding of embodied concepts. Additionally, our current benchmark only examines the

fine-grained understanding ability of specific physical concepts, while neglecting the more complex physical understanding that involves multiple interactions or observations within a single example. Developing a dataset that encompasses compositional physical concepts holds promise for future research.

**Limited Adoption of VLMs** While there are many multi-modal models available, we restrict our investigation to visual-linguistic models (VLMs) based on CLIP and its variants. We choose CLIP for its superior image representation performance and support for text-only encoders. Since our evaluation focuses on language-oriented tasks, we require models that can handle inputs consisting of pure text. Consequently, VLMs like UNITER (Chen et al., 2019), which require multi-modal inputs, are not considered. CLIP is selected as a representative work of VLMs for evaluation. However, it is important to note that the findings from CLIP may not readily generalize to other V+L models, as CLIP utilizes a large dataset of million-level image-text pairs collected from the web, which could be a significant source of embodied knowledge itself. Furthermore, there have been recent proposals for VLM models with various architectures and pre-training recipes, such as SimVLM (Wang et al., 2022), UniT (Hu and Singh, 2021), ViLT (Kim et al., 2021), and FLAVA (Singh et al., 2022), as well as vision-enhanced multi-modal agents like InstructBLIP (Dai et al., 2023), Qwen-VL (Bai et al., 2023), and Ying-VLM (Li et al., 2023). These models have shown promising performance in both cross-modal and single-modality tasks, and we look forward to evaluating these advanced models in our benchmark in the future.

## Acknowledgments

We thank all the anonymous reviewers for their constructive comments, Yuxuan Fan, Shuhuai Ren and Sijie Cheng for their valuable suggestions in preparing the manuscript.

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

# Appendix

## A  Matching-based Prompting Results for BERT

We perform the identical prompting method and templates with CLIP VLMs for LMs of BERT. RoBERTa is discarded as there is no pooled embedding during its pre-training for representing the sentence. As shown in Table 9, the performance is still close to a random guessing baseline. Besides, the higher variances on all the tasks compared to VLMs of CLIP indicate that this method does not fit LMs of BERT well, validating our prompting design choice in the main paper to fit the pre-training paradigm.

## B  Model Configurations

Here we provide the detailed configurations of evaluated LMs and VLMs in our main paper. For the vanilla model, there model configurations are listed in Table 6. For the models pre-trained from scratch

with the YFCC-15M dataset, they are adopted the same LM architecture as the VLM of CLIP-ViT/B-32, and the only difference is the pre-training objective, as shown in Table 7. All these models are optimized using an Adam optimizer with a learning rate set to $1e-4$, linearly increased at the first 2000 steps. The batch size is 2048 and all models are trained with 32 epochs. We use 1% of the data for evaluation, and the final OPT-YFCC15M model gets a 31.9 validation perplexity.

## C  Details of Prompts

We provide the used prompts for evaluating different models based on their pre-training objectives. Examples of Head, Rel and Tail of each dataset are shown in Table 1. Due to the sensitivity of language models to prompts, we provide diverse prompts for each model on each task.

**Prompts for Masked Language Models**  A `[MASK]` token is placed in the prompt and the models are asked to predict the probabilities of the `[MASK]` token. To avoid multiple mask tokens in prompts, we follow Schick and Schütze to convert knowledge fact into a cloze-question. For example, a temperature fact (`water, colder than, frying oil`) can be converted into `Q: is the water colder than frying oil? A: [MASK]!`. The models need to choose the token `yes` or `no` to fill the mask.

**Prompts for Causal Language Models**  As there is no special `[MASK]` token during the pre-training of causal language models, we do not use `[MASK]` tokens in prompts for causal language models. For Color, Shape and Material datasets of the visual concepts we construct two prompts for (Head, $\text{Tail}_1$) and (Head, $\text{Tail}_2$); while for other datasets, we construct two prompts for (Head, Rel, Tail) and (Head, Rel$'$, Tail) where Rel$'$ is the antonym relation of Rel. The prediction is based on the prompt with lower perplexity.

**Prompts for CLIP**  Following Radford et al. (2021), we use prompts like `a photo of ...` here. As the language encoder of CLIP encodes sentences to a vector and can evaluate similarities between sentences. We use an attribute prompt like `a photo of a cold object` and construct same prompts for objects (water and frying oil) in the knowledge fact. We can determine the colder object if the prompt of this object has a higher similarity to the attribute prompt.

| Model | Hidden Layers | Hidden Size | Attention Heads | Total # of Parameters |
|---|---|---|---|---|
| BERT-base | 12 | 768 | 12 | 110M |
| BERT-large | 24 | 1,024 | 16 | 340M |
| RoBERTa-base | 12 | 768 | 12 | 125M |
| RoBERTa-large | 24 | 1,024 | 16 | 355M |
| OPT-125M | 12 | 768 | 12 | 125M |
| OPT-1.3B | 24 | 2,048 | 32 | 1.3B |
| OPT-13B | 40 | 5,120 | 40 | 13B |
| OPT-175B | 96 | 12,288 | 96 | 175B |
| CLIP-ViT/B-32 | 12 | 512 | 8 | 63M |
| DeCLIP-ViT/B-32 | 12 | 512 | 8 | 63M |
| CLIP-ViT/L-14 | 12 | 768 | 12 | 123M |
| BLIP-base | 12 | 768 | 12 | 138M |

Table 6: Detailed configuration of models evaluated in the paper.

| Model | Training Objective | Training Dataset |
|---|---|---|
| BERT$_{\text{YFCC-15M}}$ | Masked Language Modeling (MLM) | Captions in YFCC-15M |
| GPT$_{\text{YFCC-15M}}$ | Causal Language Modeling (MLM) | Captions in YFCC-15M |
| CLIP$_{\text{YFCC-15M}}$ | Contrastive Image-text Matching (CIM) | Image-Text Pair in YFCC-15M |

Table 7: Pre-training objectives and corpus comparison of YFCC-15M models evaluated in the main paper.

| Model | SST-2 | QQP | QNLI | MNLI (m / mm) | Avg. |
|---|---|---|---|---|---|
| BERT (Wiki) | 90.13 | 83.20 | 87.57 | 78.90 / 80.05 | 83.97 |
| DistilledOscar | 89.33 | 67.98 | 82.48 | 74.46 / 74.82 | 77.81 |
| VLM-BERT-base | 90.60 | 90.10 | 89.47 | 81.57 / 82.43 | 86.83 |
| VLM-RoBERTa-base | 90.13 | 88.44 | 87.91 | 80.37 / 80.43 | 85.46 |

| Model | Color | Shape | Size | Height | Material | Mass | Temperature | Hardness | Avg. |
|---|---|---|---|---|---|---|---|---|---|
| BERT (Wiki) | 49.41 | 48.07 | 51.70 | 49.46 | 52.39 | 48.85 | 51.07 | 52.34 | 50.41 |
| DistilledOscar | 49.97 | 53.61 | 49.07 | 49.80 | 51.46 | 51.22 | 47.94 | 51.23 | 50.54 |
| VLM-BERT-base | 50.69 | 50.07 | 51.00 | 50.92 | 53.89 | 44.83 | 50.64 | 49.22 | 50.16 |
| VLM-RoBERTa-base | 49.53 | 51.21 | 49.00 | 49.22 | 49.54 | 49.92 | 51.11 | 49.63 | 49.90 |

Table 8: Fine-tuned accuracy of other visual-informed pre-trained language models on NLU tasks and zero-shot results regarding the physical concepts.

| Model | Color | Shape | Size | Height | Material | Mass | Temperature | Hardness | Avg. |
|---|---|---|---|---|---|---|---|---|---|
| CLIP-ViT/L-14 | $80.33_{\pm 3.61}$ | $\mathbf{85.00}_{\pm 4.03}$ | $63.96_{\pm 6.10}$ | $60.72_{\pm 5.56}$ | $80.33_{\pm 3.61}$ | $73.15_{\pm 6.34}$ | $65.88_{\pm 2.31}$ | $69.57_{\pm 2.26}$ | 72.37 |
| BERT-base Pooled | $43.19_{\pm 5.13}$ | $59.64_{\pm 7.24}$ | $66.10_{\pm 7.91}$ | $65.48_{\pm 6.63}$ | $46.55_{\pm 6.18}$ | $52.32_{\pm 7.46}$ | $56.59_{\pm 5.65}$ | $55.51_{\pm 5.50}$ | 55.67 |
| BERT-large Pooled | $44.74_{\pm 5.93}$ | $56.93_{\pm 6.45}$ | $53.80_{\pm 5.92}$ | $54.84_{\pm 8.01}$ | $52.18_{\pm 4.88}$ | $57.92_{\pm 7.26}$ | $51.90_{\pm 6.56}$ | $56.22_{\pm 4.45}$ | 53.57 |

Table 9: Zero-shot results of BERT models with pooled output as sentence embedding on VEC benchmark.

## D Entity Analysis

In our main paper, we investigate the random-level performance of BERT models by exploring the correct ratio over different prompts. We provide full histogram plots of all tasks in Figure 7, 8, 9,10, 11, 12, 13, and 14. It can be found that for visual concepts tasks such as material and shape, there are entities that BERT could produce consistent correct prediction across different prompts. However, for all embodied tasks, the histograms exhibit bell curves, indicating the poor understanding ability of BERT on embodied concepts.

## E Embodied Knowledge Transfer

We provide implementation details here for the knowledge transfer experiments from VLMs to LMs. Specifically, we take the VLM as a teacher model $T$ (e.g., the text encoder of the CLIP model) and the LM as a student model $S$ (e.g., the OPT language model). Given a text $\mathbf{x}$ from the training

dataset $\mathcal{D}$, we transfer the sequential activation patterns of $T(\mathbf{x}) \in R^{|x| \times d}$ to $S(\mathbf{x}) \in R^{|x| \times d}$, where $T(\mathbf{x})$ and $S(\mathbf{x})$ denote the last hidden representations of the VLM and the LM, respectively. $d$ is the number of hidden units. The squared maximum mean discrepancy (MMD) with kernel trick (Huang and Wang, 2017) is adopted to measure the distance between the activation patterns:

$$
\begin{aligned}
\mathrm{MMD}^2(\mathbf{x}) =& \frac{1}{d^2} \sum_{i=1}^{d} \sum_{i'=1}^{d} k \left[ S(\mathbf{x})_{*,i}; S(\mathbf{x})_{*,i'} \right] \\
&+ \frac{1}{d^2} \sum_{j=1}^{d} \sum_{j'=1}^{d} k \left[ T(\mathbf{x})_{*,j}; T(\mathbf{x})_{*,j'} \right] \\
&- \frac{2}{d^2} \sum_{i=1}^{d} \sum_{j=1}^{d} k \left[ S(\mathbf{x})_{*,i}; T(\mathbf{x})_{*,j} \right]
\end{aligned}
$$

We adopt a polynomial kernel $k(\mathbf{x}; \mathbf{y}) = \left( \mathbf{x}^\top \mathbf{y} + c \right)^p$ with $p = 2$ and $c = 0$. The MMD objective $\mathcal{L}_{\mathrm{MMD}}$ is minimized along with the original language modeling objective $\mathcal{L}_{\mathrm{LM}}$:

$$
\mathcal{L} = \mathcal{L}_{\mathrm{lm}} + \beta \mathcal{L}_{\mathrm{MMD}}
$$

where $\beta$ is a weighting factor set to 20 to achieve a balance between objectives.

## F Evaluation and Distillation with Oscar

We examine whether other vanilla distillation from traditional V+L pre-training methods brings gains regarding visual and embodied knowledge. Specifically, following Zhang et al. (2022a), we distill the knowledge of Oscar (Li et al., 2020) into a BERT model by performing knowledge distillation (Hinton et al., 2015) on the image-caption pair dataset. Specifically, the paired text and image are fed into the Oscar model for getting the vision-aware vocabulary distribution, and a student BERT model is performing masked language modeling on the text data only and learns from the soft labels provided by the Oscar teacher model. The distillation results in a DistilledOscar model supporting text-only inputs. We also evaluate VLM-BERT learned via Vokenziation (Tan and Bansal, 2020), which devises a fine-grained token-voken matching framework to utilize visual supervision. The models are evaluated on the four largest datasets in GLUE, including SST-2 (Socher et al., 2013), QQP (Iyer et al., 2017), QNLI (Rajpurkar et al., 2016) and MNLI (Williams et al., 2018) for stable results. As shown in Table 8, DistilledOscar performs worse

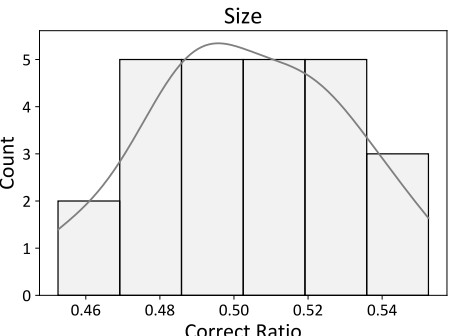

Figure 7: Histogram of entity correct ratio across different prompts on the Size dataset.

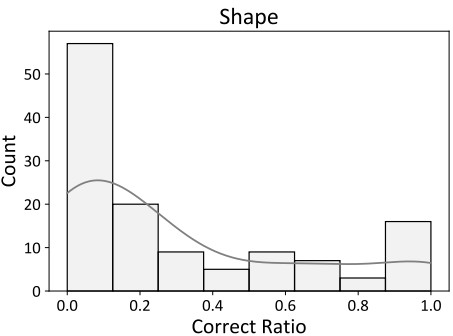

Figure 8: Histogram of entity correct ratio across different prompts on the Shape dataset.

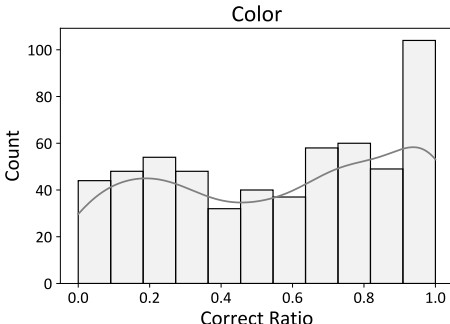

Figure 9: Histogram of entity correct ratio across different prompts on the Color dataset.

than the vanilla BERT in both NLU tasks and probing tasks regarding visual and embodied knowledge. Besides, while VLM-BERT achieves improvements on NLU tasks, it still performs at the random level on the probed tasks. These indicate that not all VLMs could learn embodied knowledge and it is non-trivial to distill the visual supervision from VLMs to LMs via purely language modeling.

Table 10: Prompts for Masked Language Models

| Model | Task | Prompt |
|---|---|---|
| BERT & RoBERTa | Size, Height, Temperature, Weight, Hardness | is the [Head] [Rel] than the [Tail]? [MASK]!
is the [Head] [Rel] than the [Tail]? [MASK].
is [Head] [Rel] than [Tail]? [MASK]!
is [Head] [Rel] than [Tail]? [MASK].
is [Head] [Rel] compared with [Tail]? [MASK].
is [Head] [Rel] compared with [Tail]? [MASK]!
compared with [Tail], is [Head] [Rel]? [MASK].
compared with [Tail], is [Head] [Rel]? [MASK]!
is [Head] usually [Rel] than [Tail]? [MASK].
is [Head] usually [Rel] than [Tail]? [MASK]! |
| | Color | can [Head] be of color [Tail]? [MASK]!
can [Head] be of color [Tail]? [MASK].
is the color of a [Head] [Tail]? [MASK]!
is the color of a [Head] [Tail]? [MASK].
is [Head] [Tail]? [MASK].
is [Head] [Tail]? [MASK]!
is [Head] typically in [Tail]? [MASK].
is [Head] typically in [Tail]? [MASK]!
Q: is [Head] of color [Tail]? A: [MASK].
Question: is [Head] of color [Tail]? Answer: [MASK]. |
| | Shape | can [Head] be the shape of [Tail]? [MASK].
can [Head] be the shape of [Tail]? [MASK]!
does the [Head] have a shape of [Tail]? [MASK].
does the [Head] have a shape of [Tail]? [MASK]!
is [Head] of [Tail]? [MASK].
is [Head] of [Tail]? [MASK]!
Q: is [Head] of [Tail]? A: [MASK].
Question: is [Head] of [Tail]? Answer: [MASK].
[Tail] [Head]? [MASK].
is [Head] typically [Tail]? [MASK]. |
| | Material | can [Head] be made of [Tail]? [MASK]!
can [Head] be made of [Tail]? [MASK].
is [Head] made of [Tail]? [MASK]!
is [Head] made of [Tail]? [MASK].
is [Tail] the necessary material for making [Head]? [MASK].
is [Tail] the necessary material for making [Head]? [MASK]!
does [Head] consist of [Tail]? [MASK].
is [Head] made up of [Tail]? [MASK].
Q: is [Head] made of [Tail]? A: [MASK].
Question: is [Head] made of [Tail]? Answer: [MASK]. |

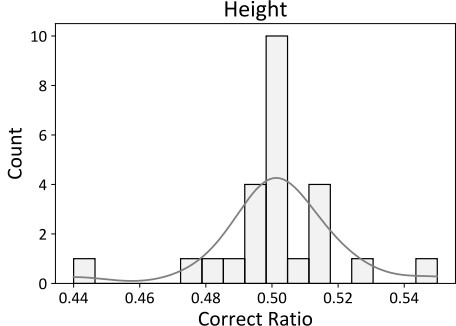

Figure 10: Histogram of entity correct ratio across different prompts on the Height dataset.

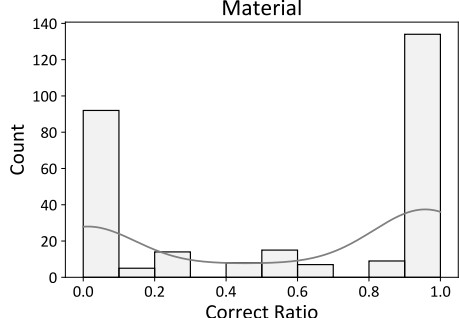

Figure 11: Histogram of entity correct ratio across different prompts on the Material dataset.

Table 11: Prompts for Causal Language Models

| Model | Task | Prompt |
|---|---|---|
| OPT | Size, Height, Temperature, Weight, Hardness | the [Head] is [Rel] than the [Tail].
[Head] is [Rel] than [Tail].
acutally, the [Head] is [Rel] than the [Tail].
acutally, [Head] is [Rel] than [Tail].
it is well-known that [Head] is [Rel] than [Tail].
[Head] is indeed [Rel] than [Tail].
the [Head] is indeed [Rel] than [Tail].
compared with the [Head], the [Tail] is [Rel].
a/(an) [Head] is [Rel] than a/(an) [Tail].
yes, [Head] is [Rel] than [Tail]. |
| | Color | [Head] can be of the color [Tail].
the [Head] can be of color [Tail].
the color of a(an) [Head] is [Tail].
the color of [Head] is [Tail].
the [Head] is in [Tail].
[Head] is [Tail].
what color is the [Head]? [Tail].
[Head]'s color is [Tail].
usually, [Head] is in [Tail].
[Head] is typically [Tail]. |
| | Shape | [Head] is usually [Tail].
what is the shape of [Head]? [Tail].
[Head] is typically [Tail].
[Head]'s shape is [Tail]. |
| | Material | [Head] is made of [Tail].
the [Head] is made of [Tail].
[Head] consists of [Tail].
the main material of [Head] is [Tail].
[Tail] is necessary material for making [Head].
the [Head] consists of [Tail].
the [Head] can be made of [Tail].
the [Head] is built with [Tail].
the [Head] contains [Tail].
the [Head] is made up of [Tail]. |

Table 12: Prompts used for VLMs of CLIP.

| Model | Task | Prompt |
|---|---|---|
| CLIP | All Tasks | a photo of a [Head]/[Attribute].
a photo of the [Head]/[Attribute].
a blurry photo of a [Head]/[Attribute].
a good photo of a [Head]/[Attribute].
a painting of a [Head]/[Attribute].
a bad photo of a [Head]/[Attribute].
a close-up photo of a [Head]/[Attribute].
a bright photo of the [Head]/[Attribute].
a photo of one [Head]/[Attribute].
a low resolution photo of a [Head]/[Attribute]. |

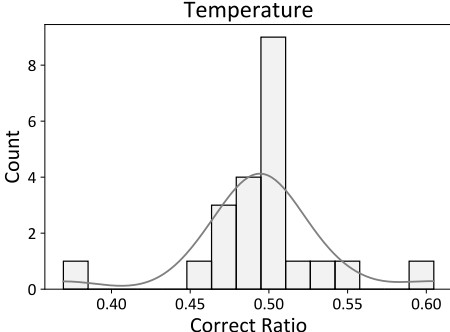
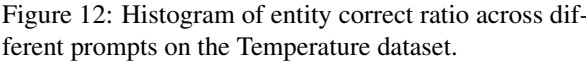

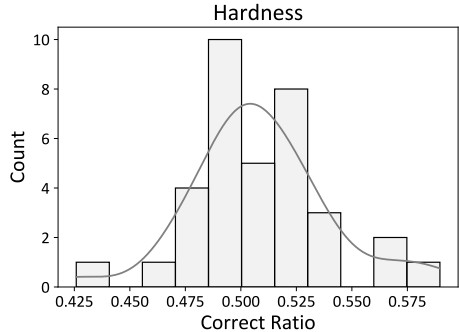

Figure 12: Histogram of entity correct ratio across different prompts on the Temperature dataset.

Figure 13: Histogram of entity correct ratio across different prompts on the Hardness dataset.

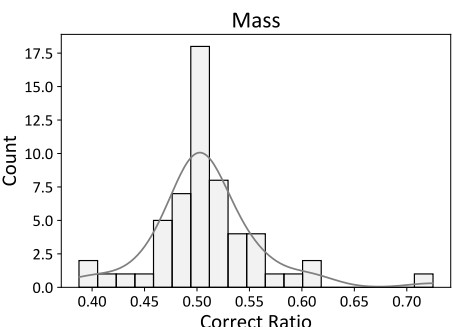

Figure 14: Histogram of entity correct ratio across different prompts on the Mass dataset.