# OpenReview forum: "Can Language Models Understand Physical Concepts?"
_EMNLP/2023/Conference — EMNLP 2023 Main_

### Official Review · Reviewer_MVNy · 2023-07-28

**Soundness:** 4

**Excitement:**

4: Strong: This paper deepens the understanding of some phenomenon or lowers the barriers to an existing research direction.

**Paper Topic And Main Contributions:**

The paper investigated language models' understanding of physical concepts and designs a benchmark called VEC. The main contributions of this paper:

- conducted in-depth analysis with different types of LLMs and VLMs on their understanding of visual concepts and embodied concepts;
- provided insights on what kind of physical concepts are well-captured and struggled by LLMs; showing evidence that vision-language pretraining helps learn embodied knowledge
- showed that distilling knowledge from VLM to LLM significantly boost performance on understanding embodied concepts

**Questions For The Authors:**

Please see the sub-bullets in the Reasons to reject.

**Reasons To Accept:**

- The proposed benchmark is a good combination of previous word and newly collected datasets, covering both visual and embodied concepts;
- The experiments are solid with clear findings and insights. The finding that we can effectively distill embodied knowledge from VLMs to less capable LLMs is quite inspiring;
- The paper is well-written and easy-to-follow;

**Reasons To Reject:**

- Lacking a comparison or case study on recent LLMs (models with instruction-following SFT or RLHF), such as ChatGPT, GPT-4, and Llama-family.

  - Based on my own observations, when compared with GPT-3 era models (analyzed in this paper), newer models like ChatGPT contain much richer physical commonsense knowledge and can perform complex reasoning about physical interactions. As a result, some of the conclusions presented in this paper may not hold true for these newer models. A particular question that I expected to see in this paper was: to what extent can these pure-text models understand the physical world? Furthermore, what causes the performance boost in these newer models if scaling is not the sole reason? (Could it be RLHF, and if so, why?)

- The evaluation setting for CLIP is quite different from BERT and OPT, which might be argued whether it is an apples-to-apples comparison.
    - If I understand correctly, the matching-based method for CLIP essentially compares the level of correlation between an object and an attribute instead of directly comparing two objects.
    - A smaller cosine score between "This is a photo of the water" and "Attribute: This is a photo of a cold object" versus "This is a photo of frying oil" can only indicate that "water" is more likely to be associated or co-occur with "cold" compared to "oil"; however, can we say that the model understands "water" is "colder" than "frying oil"?

**Reproducibility:**

5: Could easily reproduce the results.

**Reviewer Confidence:**

4: Quite sure. I tried to check the important points carefully. It's unlikely, though conceivable, that I missed something that should affect my ratings.

---

> ### Author Rebuttal · Authors · 2023-08-28
>
> We deeply appreciate your thoughtful and constructive review. Your recognition of our benchmark's combination of existing and new datasets, as well as your acknowledgment of our solid experiments and clear findings, is greatly encouraging. We're also pleased that you found our paper well-written and easy to follow. Thank you for your time and valuable inputs, and the responses are provided below.
>
> ## Evaluation Results with Recent LLMs
>
> We supplement the results with LLaMa-1/2, Vicuna and LLaMa-2-Chat for a better understanding the scaling and the effect of SFT and RLHF. Commercial APIs such as ChatGPT and GPT-4 are excluded to avoid potential data contamination due to the intransparent usage of user data.
>
>
> **Results on Visual Datasets**
>
> | Model           | Color           | Shape          | Size            | Height         | Material       | Average |
> | --------------- | --------------- | -------------- | --------------- | -------------- | -------------- | ------- |
> | LLaMa-1-7B     | 63.94  $\pm$4.87   | 66.19  $\pm$2.36  | 65.91  $\pm$9.86   | 50.00  $\pm$0.00  | 66.76  $\pm$3.88  | 62.56   |
> | Vicuna-7B-v1.3  | 64.31  $\pm$5.44   | 73.33  $\pm$2.88  | 62.50  $\pm$8.80   | 50.02  $\pm$0.11  | 68.31  $\pm$3.75  | 63.69   |
> | LLaMa-1-13B   | 66.27  $\pm$3.89   | 62.38  $\pm$2.36  | 63.14 $\pm$11.06   | 50.16  $\pm$0.43  | 65.46  $\pm$2.95  | 61.48   |
> | Vicuna-13B-v1.3 | 66.11  $\pm$5.62   | 67.38  $\pm$2.69  | 64.35  $\pm$13.62  | 50.92  $\pm$2.53  | 68.52  $\pm$5.60  | 63.46   |
> | LLaMa-2-7B     | 63.73 $\pm$ 3.09 | 65.24 $\pm$ 4.22 | 61.88 $\pm$ 9.37 | 50.02 $\pm$ 0.06 | 66.34 $\pm$ 3.25 | 61.44 |
> | LLaMa-2-7B Chat | 60.99 $\pm$ 5.18 | 70.95 $\pm$ 2.76 | 68.03  $\pm$ 9.91 | 51.72  $\pm$ 2.20 | 67.39  $\pm$ 4.13 | 63.82 |
> | LLaMa-2-13B     | 66.59 $\pm$  3.40 | 62.38 $\pm$ 3.21 | 68.20  $\pm$ 11.51 | 50.10  $\pm$ 0.18 | 66.73 $\pm$   3.99 | 62.80 |
> | LLaMa-2-13B Chat | 64.04 $\pm$ 4.41 | 70.71$\pm$ 1.75 | 70.68 $\pm$  8.79 | 51.18 $\pm$  1.61 | 67.96  $\pm$ 4.63 | 64.91 |
> | OPT-13B | 79.62  $\pm$5.28 | 62.50 $\pm$6.44 | 57.56 $\pm$6.60 | 54.58 $\pm$4.53 | 88.38 $\pm$3.14 | 68.53 |
> | BLIP | 82.60 $\pm$5.50 | 84.86 $\pm$2.80 | 76.00 $\pm$6.40 | 69.84 $\pm$7.76 | 80.67  $\pm$1.24 | 78.79 |
>
>
>
> **Results on Embodied Datasets**
>
> | Model           | Mass           | Temperature    | Hardness       | Average |
> | --------------- | -------------- | -------------- | -------------- | ------- |
> | LLaMa-1-7B      | 54.88  $\pm$2.49  | 60.69  $\pm$4.35  | 51.97  $\pm$2.84  | 55.84   |
> | Vicuna-7B-v1.3  | 54.23  $\pm$1.78  | 58.85  $\pm$4.36  | 54.42  $\pm$6.42  | 55.83 |
> | LLaMa-1-13B   | 53.69  $\pm$3.81  | 50.76  $\pm$8.69  | 53.94  $\pm$4.45  | 52.80 |
> | Vicuna-13B-v1.3 | 56.90  $\pm$3.53  | 53.32  $\pm$6.47  | 55.50  $\pm$5.73  | 55.24   |
> | LLaMa-2 7B       | 54.01 $\pm$4.47  | 56.87  $\pm$6.25  | 55.22   $\pm$5.89  | 55.37   |
> | LLaMa-2 7B Chat  | 52.51 $\pm$4.83  | 61.99  $\pm$3.93  | 55.65  $\pm$  5.28 | 56.72   |
> | LLaMa-2 13B | 53.38$\pm$ 2.10  | 57.54  $\pm$ 7.51 | 53.01 $\pm$ 4.57   | 54.64   |
> | LLaMa-2 13B Chat | 54.13 $\pm$2.73  | 56.68 $\pm$ 6.02  | 54.12  $\pm$ 4.64  | 54.98   |
> | OPT-13B | 50.14  $\pm$0.36 | 51.85  $\pm$6.34 | 52.38  $\pm$3.09 | 51.46 |
> | BLIP | 83.94 $\pm$ 2.59 | 74.98 $\pm$ 5.60 | 56.93 $\pm$ 5.56 | 71.95 |
>
> Based on the updated results, we provide point-to-point response for the raised questions.
>
> **Question: Do the results hold for newer models?**
>
> **Response**:
>
> We find that OPT-13B and LLaMa 13B excel in understanding different visual concepts, and LLaMa series models perform slightly better regarding the embodied concepts. However, there is still a clear performance gap between the text-only LLMs and VLMs (such as BLIP). Besides, there are specific tasks, such as relative height recognition, that model scaling does not help. These findings are consistent with our previous investigation with OPT-series models and show that pure-text models still struggle with physical concepts, especially for the embodied concepts.
>
> **Question: To what extent can these pure-text models understand the physical world?**
>
> **Response**: Based on our updated results, pure-text models demonstrate a commendable understanding of visual concepts, including aspects such as relative size comparison and material recognition. They also exhibit an introductory level of comprehension of embodied concepts like temperature, as evidenced by an approximate 60% accuracy rate in temperature understanding tasks.
>
> **Question: What causes the performance boost in these newer models if scaling is not the sole reason? (Could it be RLHF, and if so, why?)**
>
> **Response**: We have observed that Vicuna models, after undergoing instruction tuning with SFT data, exhibit enhanced performance in understanding physical concepts. This improvement is particularly noticeable in larger language models. For instance, when compared to the base LLaMa 13B model, the average embodied accuracy of Vicuna-13B increases from 52.80 to 55.24. This suggests that a combination of instruction tuning and model size scaling could contribute to the performance boost.
>
> As for the role of RLHF, while LLaMa-2-chat consistently outperforms LLaMa-2, the gap is not substantial enough to conclusively attribute the improvement to RLHF. Given the absence of LLMs solely incorporating RLHF, we leave a more comprehensive investigation of RLHF's impact for future work.
>
>
>
>
> ## Evaluation Setup of CLIP
>
> **Question: On the Evaluation Difference Between VLMs (i.e., CLIP) and PLMs (BERT, OPT)**
>
> **Response**: Yes, your understanding of the matching-based method is accurate. As outlined in lines 225-228 of our paper, all prompting methods are tailored to best fit the respective pre-training paradigms, a strategy recent studies have corroborated as effective. Given this, we argue that our comparison is fair, as these prompting methods all align with the pre-training objectives used to extract learned knowledge from the pre-training phase. Furthermore, we applied matching-based prompting methods to BERT's pooled embedding, as referenced in Appendix A and noted in the footnote on Page 4. Our findings indicate that employing a matching-based method for BERT yields marginally superior results but with greater variances across prompts. However, a significant performance gap between CLIP and BERT persists, aligning with our main paper's results.
>
>
> **Question: On the correlation between the cosine similarity and understanding of specific concepts**
>
> **Response**: Thank you for posing such as interesting question. To begin, it's key to emphasize that both CLIP and language models like BERT or GPT fundamentally operate as statistical models. They generate predictions based on the patterns they've learned from their training data, thereby mimicking a form of understanding that is more about recognizing these patterns than exhibiting true comprehension in the human sense. This understanding ability is demonstrated by co-occurrence relationships between objects. For instance, if `water` frequently co-occurs with `cold` and `frying oil` does so less often, indicated by the cosine similarities between sentences, the model might infer a relative temperature difference based on these associations. To verify this, we compute the text embeddings for `a photo of water` and ` a photo of frying oil` with the text encoder of CLIP-ViT-L/14, coupled with a set of temperature sentences describing `a photo of an object at X degrees Celsius`. Here, `X` represents a range of common temperatures including `[0, 10, 20, 50, 100, 200]`. We find that the sentence representation of the water photo most closely aligns with 10 degrees Celsius, while the sentence representation for the frying oil photo is closest to 50 degrees Celsius. This outcome suggests that the model uses co-occurrence patterns to infer a relative temperature difference between objects.
>
>
>
>
> We hope these explanations clarify our evaluation setup and findings, and we would gladly incorporate this discussion in our revision.

---

### Official Review · Reviewer_kY1c · 2023-08-03

**Soundness:** 4

**Excitement:**

4: Strong: This paper deepens the understanding of some phenomenon or lowers the barriers to an existing research direction.

**Paper Topic And Main Contributions:**

This paper proposes an evaluation suite of physical concepts covering visual and embodied concepts. They probe text-only LMs and VLMs in this benchmark. They find that the understanding of certain visual concepts emerges as scaling up LMs, and vision-augmented LMs can gain better performance compared with text-only LMs. Based on this, they propose a distillation method to transfer embodied knowledge from VLMs to LMs to gain better performance.

**Reasons To Accept:**

- Probing LM with physical concepts is a novel and interesting topic, it provides insights of how to further improve and align current LMs (especially LLMs) to real world.

- The task format is well-designed, and the two kinds of concepts are defined clearly.

- probing methods for different LMs and VLM are well-designed, and the experiments are solid.

**Reasons To Reject:**

- The models used in this paper mainly are OPT and Encoder-only LMs, I think it would be more interesting and meaningful to explore more recent state-of-the-art LLMs such as ChatGPT and Llama, due to current LLMs are aligned with human instructions, they might can gain better or near to human performance.

**Reproducibility:**

4: Could mostly reproduce the results, but there may be some variation because of sample variance or minor variations in their interpretation of the protocol or method.

**Reviewer Confidence:**

4: Quite sure. I tried to check the important points carefully. It's unlikely, though conceivable, that I missed something that should affect my ratings.

---

> ### Author Rebuttal · Authors · 2023-08-28
>
> Thank you for your valuable feedback. We appreciate your recognition of our interesting topic, well-designed task format, and solid experimental results.
>
> In response to your comment suggesting exploring recent LLMs, we would like to provide additional information regarding the evaluation results. We select LLaMa-1/2 (7B and 13B) as a representative of well-performing LLMs, Vicuna (v1.3, 7B and 13B) to explore the effect of instruction tuning, and LLaMa-2 Chat is also incorporated for exploring the impact of RLHF. We exclude commercial LLM APIs such as ChatGPT because of their intransparency and the potential issue of data contamination.
>
>
> **Results on Visual Datasets**
>
> | Model           | Color           | Shape          | Size            | Height         | Material       | Average |
> | --------------- | --------------- | -------------- | --------------- | -------------- | -------------- | ------- |
> | LLaMa-1-7B     | 63.94  $\pm$4.87   | 66.19  $\pm$2.36  | 65.91  $\pm$9.86   | 50.00  $\pm$0.00  | 66.76  $\pm$3.88  | 62.56   |
> | Vicuna-7B-v1.3  | 64.31  $\pm$5.44   | 73.33  $\pm$2.88  | 62.50  $\pm$8.80   | 50.02  $\pm$0.11  | 68.31  $\pm$3.75  | 63.69   |
> | LLaMa-1-13B   | 66.27  $\pm$3.89   | 62.38  $\pm$2.36  | 63.14 $\pm$11.06   | 50.16  $\pm$0.43  | 65.46  $\pm$2.95  | 61.48   |
> | Vicuna-13B-v1.3 | 66.11  $\pm$5.62   | 67.38  $\pm$2.69  | 64.35  $\pm$13.62  | 50.92  $\pm$2.53  | 68.52  $\pm$5.60  | 63.46   |
> | LLaMa-2-7B     | 63.73 $\pm$ 3.09 | 65.24 $\pm$ 4.22 | 61.88 $\pm$ 9.37 | 50.02 $\pm$ 0.06 | 66.34 $\pm$ 3.25 | 61.44 |
> | LLaMa-2-7B Chat | 60.99 $\pm$ 5.18 | 70.95 $\pm$ 2.76 | 68.03  $\pm$ 9.91 | 51.72  $\pm$ 2.20 | 67.39  $\pm$ 4.13 | 63.82 |
> | LLaMa-2-13B     | 66.59 $\pm$  3.40 | 62.38 $\pm$ 3.21 | 68.20  $\pm$ 11.51 | 50.10  $\pm$ 0.18 | 66.73 $\pm$   3.99 | 62.80 |
> | LLaMa-2-13B Chat | 64.04 $\pm$ 4.41 | 70.71$\pm$ 1.75 | 70.68 $\pm$  8.79 | 51.18 $\pm$  1.61 | 67.96  $\pm$ 4.63 | 64.91 |
> | OPT-13B | 79.62  $\pm$5.28 | 62.50 $\pm$6.44 | 57.56 $\pm$6.60 | 54.58 $\pm$4.53 | 88.38 $\pm$3.14 | 68.53 |
> | BLIP | 82.60 $\pm$5.50 | 84.86 $\pm$2.80 | 76.00 $\pm$6.40 | 69.84 $\pm$7.76 | 80.67  $\pm$1.24 | 78.79 |
>
>
>
>
>
> **Results on Embodied Datasets**
>
> | Model           | Mass           | Temperature    | Hardness       | Average |
> | --------------- | -------------- | -------------- | -------------- | ------- |
> | LLaMa-1-7B      | 54.88  $\pm$2.49  | 60.69  $\pm$4.35  | 51.97  $\pm$2.84  | 55.84   |
> | Vicuna-7B-v1.3  | 54.23  $\pm$1.78  | 58.85  $\pm$4.36  | 54.42  $\pm$6.42  | 55.83 |
> | LLaMa-1-13B   | 53.69  $\pm$3.81  | 50.76  $\pm$8.69  | 53.94  $\pm$4.45  | 52.80 |
> | Vicuna-13B-v1.3 | 56.90  $\pm$3.53  | 53.32  $\pm$6.47  | 55.50  $\pm$5.73  | 55.24   |
> | LLaMa-2 7B       | 54.01 $\pm$4.47  | 56.87  $\pm$6.25  | 55.22   $\pm$5.89  | 55.37   |
> | LLaMa-2 7B Chat  | 52.51 $\pm$4.83  | 61.99  $\pm$3.93  | 55.65  $\pm$  5.28 | 56.72   |
> | LLaMa-2 13B | 53.38$\pm$ 2.10  | 57.54  $\pm$ 7.51 | 53.01 $\pm$ 4.57   | 54.64   |
> | LLaMa-2 13B Chat | 54.13 $\pm$2.73  | 56.68 $\pm$ 6.02  | 54.12  $\pm$ 4.64  | 54.98   |
> | OPT-13B | 50.14  $\pm$0.36 | 51.85  $\pm$6.34 | 52.38  $\pm$3.09 | 51.46 |
> | BLIP | 83.94 $\pm$ 2.59 | 74.98 $\pm$ 5.60 | 56.93 $\pm$ 5.56 | 71.95 |
>
> Our key findings are:
>
> - **OPT-13B and LLaMa 13B models each excel in different visual concepts**, with OPT-13B performing well on material concepts and LLaMa 13B on relative size comparisons. This is likely due to the pre-training corpus distribution. LLaMa models also show better performance in embodied concepts, with ~60% accuracy on certain tasks.
> - **Instruction tuning improves the performance of Vicuna models in both visual and embodied concepts**, with more significant gains for larger LLMs. For instance, the average accuracy in three embodied tasks improves from 52.8 to 55.2 when using LLaMa 13B as a baseline.
> - LLaMa-2-Chat models, further **trained with a supervised instruction tuning dataset and RLHF techniques, show consistent accuracy gains** in both visual and embodied concept tasks. However, the individual effects of instruction tuning and RLHF are unclear as they are implemented together.
> - **A clear performance gap still remains between more recent instruction-tuned and RLHF LLMs, and VLMs. However, in certain tasks like object hardness, LLaMa-2-7B performs comparably to the powerful BLIP VLM**.
>
> We will incorporate these updated results to provide a more aligned investigation with recent progress of LLMs.

---

### Official Review · Reviewer_zxLC · 2023-08-11

**Typos Grammar Style And Presentation Improvements:** Some minor typos but overall good pre…
**Soundness:** 4

**Excitement:**

4: Strong: This paper deepens the understanding of some phenomenon or lowers the barriers to an existing research direction.

**Missing References:**

No

**Paper Topic And Main Contributions:**

This paper examines the capacity of language models (LMs) to comprehend physical concepts in the real world. The authors create the VEC benchmark, encompassing visual and embodied attributes. Visual concepts encompass properties like color, shape, and material, while embodied concepts involve mass, temperature, and hardness. The study evaluates different LMs, including text-based ones like BERT and GPT (OPT)-family, as well as vision-augmented models like CLIP and BLIP.

The paper's contributions are as follows:

**VEC Benchmark:** Introducing the VEC benchmark enables the assessment of LMs' grasp of physical attributes, covering both visual and embodied concepts.

**Understanding with Scaling:** The findings reveal that as LMs scale, they gain better understanding of certain visual concepts. However, understanding embodied knowledge remains a challenge, despite performing better than random guessing. This underscores the need for advancing embodied concept comprehension.

**Vision-Augmented LMs:** Vision-augmented models like CLIP and BLIP achieve human-level comprehension of embodied concepts, indicating the significance of visual data in understanding physical attributes.

**Knowledge Distillation:** Inspired by vision-augmented LMs, a knowledge distillation method is proposed, transferring embodied knowledge to standard LMs. This yields performance gains comparable to extensive parameter scaling.

In summary, the paper investigates LMs' grasp of physical concepts, demonstrating improved comprehension of visual concepts with scaling. However, challenges persist in understanding embodied concepts. Vision-augmented LMs exhibit promise, and a knowledge distillation approach is proposed to bridge the gap between standard and vision-augmented LMs' comprehension of physical attributes.

**Questions For The Authors:**

The questions are mostly the same as the weakness of this paper. I think addressing these weaknesses will help increase the impact of this paper.

**Reasons To Accept:**

1. The paper introduces a novel benchmark evaluation suite (VEC) that covers a wide range of physical concepts, including both visual and embodied attributes. This benchmark fills a gap in the field by providing a standardized framework to evaluate language models' understanding of physical attributes, which is essential for advancing research in this area.

2. The paper conducts a thorough evaluation of various types of language models, including text-only LMs like BERT and VL models like CLIP and BLIP.

3. This direction is very relevant as next step for grounded LMs or LLMs.

4. The paper addresses many intriguing properties of LMs including its physical ability or understanding of embodiment ab. It is interesting to see that *"the ability of certain visual concepts emerges as scaling up LMs, but there are still basic visual concepts where the scaling law fails"*, which gives strong motivation for grounded LMs.

**Reasons To Reject:**

1. The paper might have benefited from testing on open-source LLMs such as llama or instruction tuned models, but this could be attributed to the rapidly evolving landscape of LLMs and the timeline of the experiments of this paper. Given the dynamic nature of this field, including open-source LLMs could have enhanced the paper's relevance to the community and ensured its alignment with the latest developments. Still, it is worth acknowledging the largest model scale used is OPT(175B).
2. The paper can benefit from allocating more paragraphs to how they develop the dataset, which seems to be the core contribution of this paper. Moreover, it is intuitively ideal to develop a more complex embodied dataset in which a single example may involve a series of embodied or physical interaction/observation. Current paper usually addresses single embodied attribute like mass/temperature.

**Reproducibility:**

5: Could easily reproduce the results.

**Reviewer Confidence:**

4: Quite sure. I tried to check the important points carefully. It's unlikely, though conceivable, that I missed something that should affect my ratings.

---

> ### Author Rebuttal · Authors · 2023-08-28
>
> We deeply appreciate your insightful comments and the recognition you have given to our work. Your positive assessment of the VEC benchmark, experimental design, findings, and knowledge distillation exploration is truly inspiring. We are grateful for your valuable feedback, and we have provided detailed responses to your comments below.
>
>
> ## Evaluation results with LLaMa-1/2, Vicuna and LLaMa-2-Chat
>
> Thanks for acknowledging the results obtained from the OPT-175B model, and we are happy to incorporate more results with recent LLMs.
>
> We compare the performance of LLaMA 1/2 models (7B and 13B), Vicuna models (7B, 13B, v1.3) trained with the instruction tuning dataset, and LLaMa-2 Chat models (7B and 13B) trained with SFT and RLHF, using the same setup as OPT models, as they are all casual language models.
>
> **Results on Visual Datasets**
>
> | Model           | Color           | Shape          | Size            | Height         | Material       | Average |
> | --------------- | --------------- | -------------- | --------------- | -------------- | -------------- | ------- |
> | LLaMa-1-7B     | 63.94  $\pm$4.87   | 66.19  $\pm$2.36  | 65.91  $\pm$9.86   | 50.00  $\pm$0.00  | 66.76  $\pm$3.88  | 62.56   |
> | Vicuna-7B-v1.3  | 64.31  $\pm$5.44   | 73.33  $\pm$2.88  | 62.50  $\pm$8.80   | 50.02  $\pm$0.11  | 68.31  $\pm$3.75  | 63.69   |
> | LLaMa-1-13B   | 66.27  $\pm$3.89   | 62.38  $\pm$2.36  | 63.14 $\pm$11.06   | 50.16  $\pm$0.43  | 65.46  $\pm$2.95  | 61.48   |
> | Vicuna-13B-v1.3 | 66.11  $\pm$5.62   | 67.38  $\pm$2.69  | 64.35  $\pm$13.62  | 50.92  $\pm$2.53  | 68.52  $\pm$5.60  | 63.46   |
> | LLaMa-2-7B     | 63.73 $\pm$ 3.09 | 65.24 $\pm$ 4.22 | 61.88 $\pm$ 9.37 | 50.02 $\pm$ 0.06 | 66.34 $\pm$ 3.25 | 61.44 |
> | LLaMa-2-7B Chat | 60.99 $\pm$ 5.18 | 70.95 $\pm$ 2.76 | 68.03  $\pm$ 9.91 | 51.72  $\pm$ 2.20 | 67.39  $\pm$ 4.13 | 63.82 |
> | LLaMa-2-13B     | 66.59 $\pm$  3.40 | 62.38 $\pm$ 3.21 | 68.20  $\pm$ 11.51 | 50.10  $\pm$ 0.18 | 66.73 $\pm$   3.99 | 62.80 |
> | LLaMa-2-13B Chat | 64.04 $\pm$ 4.41 | 70.71$\pm$ 1.75 | 70.68 $\pm$  8.79 | 51.18 $\pm$  1.61 | 67.96  $\pm$ 4.63 | 64.91 |
> | OPT-13B | 79.62  $\pm$5.28 | 62.50 $\pm$6.44 | 57.56 $\pm$6.60 | 54.58 $\pm$4.53 | 88.38 $\pm$3.14 | 68.53 |
> | BLIP | 82.60 $\pm$5.50 | 84.86 $\pm$2.80 | 76.00 $\pm$6.40 | 69.84 $\pm$7.76 | 80.67  $\pm$1.24 | 78.79 |
>
>
>
> **Results on Embodied Datasets**
>
> | Model           | Mass           | Temperature    | Hardness       | Average |
> | --------------- | -------------- | -------------- | -------------- | ------- |
> | LLaMa-1-7B      | 54.88  $\pm$2.49  | 60.69  $\pm$4.35  | 51.97  $\pm$2.84  | 55.84   |
> | Vicuna-7B-v1.3  | 54.23  $\pm$1.78  | 58.85  $\pm$4.36  | 54.42  $\pm$6.42  | 55.83 |
> | LLaMa-1-13B   | 53.69  $\pm$3.81  | 50.76  $\pm$8.69  | 53.94  $\pm$4.45  | 52.80 |
> | Vicuna-13B-v1.3 | 56.90  $\pm$3.53  | 53.32  $\pm$6.47  | 55.50  $\pm$5.73  | 55.24   |
> | LLaMa-2 7B       | 54.01 $\pm$4.47  | 56.87  $\pm$6.25  | 55.22   $\pm$5.89  | 55.37   |
> | LLaMa-2 7B Chat  | 52.51 $\pm$4.83  | 61.99  $\pm$3.93  | 55.65  $\pm$  5.28 | 56.72   |
> | LLaMa-2 13B | 53.38$\pm$ 2.10  | 57.54  $\pm$ 7.51 | 53.01 $\pm$ 4.57   | 54.64   |
> | LLaMa-2 13B Chat | 54.13 $\pm$2.73  | 56.68 $\pm$ 6.02  | 54.12  $\pm$ 4.64  | 54.98   |
> | OPT-13B | 50.14  $\pm$0.36 | 51.85  $\pm$6.34 | 52.38  $\pm$3.09 | 51.46 |
> | BLIP | 83.94 $\pm$ 2.59 | 74.98 $\pm$ 5.60 | 56.93 $\pm$ 5.56 | 71.95 |
>
> - Comparison of LLaMA series models with OPT-13B: Our observations reveal that **OPT-13B and LLaMa 13B models demonstrate proficiency in understanding distinct visual concepts**. OPT-13B shines in the material concept, whereas LLaMa 13B performs better in relative size comparison tasks. This discrepancy may stem from the distribution of the pre-training corpus, as evidenced by the noticeable accuracy improvement after training on the YFCC-15M visual-grounded text corpus. Furthermore, **LLaMa models exhibit better performance in embodied concepts**, manifesting an initial understanding of specific concepts, such as a ~60% accuracy rate on the temperature task of LLaMa-1-7B.
>
> - Impact of Instruction Tuning: Post-training with the instruction tuning dataset, **Vicuna models display enhanced proficiency in both visual and embodied concepts**, with larger LLMs demonstrating a more significant improvement. For instance, when using LLaMa 13B as a baseline model, the average accuracy in three embodied tasks rises from 52.8 to 55.2. This encouraging result suggests an alternative approach to enhancing embodied understanding.
>
> - Comparison of LLaMa-2 and LLaMa-2-Chat models: Chat models, further trained with the supervised instruction tuning dataset and RLHF techniques, consistently show accuracy improvements in both visual and embodied concept tasks. However, disentangling the influence of instruction tuning and RLHF on these models presents a challenge as they are intertwined.
>
> - Performance gap between instruction-tuned, RLHF LLMs and VLMs: **A clear performance disparity still exists between recent LLMs incorporating instruction tuning/RLHF and VLMs**. Nonetheless, we observed that in certain tasks, such as object hardness, LLaMa-2-7B performs comparably to the powerful BLIP VLM.
>
> We will integrate these revised results to align better with the ongoing advancements in LLM research.
>
>
> ### Details of Dataset Construction
>
>
> We appreciate your insightful feedback. To address your suggestion, we are keen to expand our dataset details in the paper.
>
> In terms of visual datasets, we've chosen the Size and Height tasks from the Spatial Commonsense [1] dataset. These tasks delve into the relative spatial relationships among 25 everyday objects. Additionally, we've included the Material, Color, and Shape tasks from the ViComTe dataset [2]. We primarily utilize the test split of each task to evaluate the LLMs.
>
> Concerning embodied datasets, we gather annotations for objects with attributes of interest from academic databases or Wikipedia. This approach yields 56, 22, and 25 different objects in the mass, temperature, and hardness datasets, respectively. Pairs of objects are formed by establishing a significant gap for the particular attribute.
>
> In our revised version, we plan to provide these extended dataset details and offer more statistics, including the distribution of objects, to portray a more comprehensive picture of the dataset.
>
>
>
> ### Developing Instances involving Compositional Physical Concepts
>
> We appreciate your comment regarding the potential development of a more complex embodied dataset that incorporates multiple physical interactions or observations within a single example.
>
> Our research primarily aims to examine the fine-grained understanding ability of specific embodied concepts, such as mass and temperature, which we observed large language models struggle with.
>
> We also agree that developing a dataset encompassing compositional physical concepts very interesting and look forward to exploring this avenue in future research.
>
>
>
> [1] Things not Written in Text: Exploring Spatial Commonsense from Visual Signals, ACL 2022
>
> [2] Visual Commonsense in Pretrained Unimodal and Multimodal Models, NAACL 2022

---

### Meta-Review · Area_Chair_XFo1 · 2023-09-18

**Recommendation:** 5

**Metareview:**

This paper proposes a new evaluation benchmark for of the ability for language models to represent real-world physical concepts, including color, shape, material, mass, temperature, hardness, etc. The paper also performs thorough evaluation of existing text-only and vision-and-language models, including (in the rebuttal) updated results with more recent models that became available after the submission deadline. The paper finds that both scaling and knowledge distillation from vision-language to text-only models improve performance for visual concepts (but not embodied concepts), and that models that use visual features perform better than text-only models. The more recent models also perform better than those available at submission time.

---

### Decision · Program_Chairs · 2023-10-07

**Decision:**

Accept-Main

**Comment:**

This paper proposes a new evaluation benchmark for of the ability for language models to represent real-world physical concepts, including color, shape, material, mass, temperature, hardness, etc. The paper also performs thorough evaluation of existing text-only and vision-and-language models, including (in the rebuttal) updated results with more recent models that became available after the submission deadline. The paper finds that both scaling and knowledge distillation from vision-language to text-only models improve performance for visual concepts (but not embodied concepts), and that models that use visual features perform better than text-only models. The more recent models also perform better than those available at submission time.